# Deep Progressive Training: scaling up depth capacity of zero/one-layer models

## Abstract

Model depth is a double-edged sword in deep learning: deeper models achieve higher accuracy but require higher computational cost. To efficiently train models at scale, progressive training – an effective strategy where model capacity scales up during training, has emerged to significantly reduce computation with little to none performance degradation. In this work, we study the depth expansion of large-scale models through the lens of optimization theory and feature learning, offering insights on the initialization of new layers, hyperparameter transfer, learning rate schedule, and timing of model expansion. Specifically, we propose zero/one-layer progressive training for the optimal tradeoff between computation and loss. For example, zero/one-layer progressive training on GPT2 can save $\approx 80\%$ compute, or equivalently accelerate by $\approx 5\times$, and achieve a loss comparable to a fully trained 60-layer model with 7B parameters.

## 1 Introduction

Strong performance of deep learning models is highly correlated to model sizes, with larger model having higher accuracy but also incurring higher computation cost to train, e.g. LLAMA-4 training costs over 7M GPU hours and an estimated 2,000 tons of carbon emissions. This phenomenon leads to a tradeoff between model utility (measured by loss or accuracy) and computational cost (measured by floating point operation, or FLOP), and has motivated scaling laws to train compute-optimal large language models (Hoffmann et al., 2022; Kaplan et al., 2020).

To accelerate the training of large models, one direction is known as progressive training, or model growth, or model reuse, which initially trains a small model (a.k.a. teacher or source model) and then scales up to large models (a.k.a. student or grown model) during training. In contrast to the fixed-size training, the progressive training formulates the model size as a time-dependent variable, and it is clearly more efficient because the compute is $6BTN$, proportional to the model size $N$. For example, consider a progressive training that scales up the model size at iteration $\tau$:

$$N(t) = \begin{cases} N_{\text{small}} & \text{if } t \leq \tau \\ N_{\text{large}} & \text{if } t > \tau \end{cases} \tag{1.1}$$

The fixed-size training requires $6BTN_{\text{large}}$ FLOPs, whereas the progressive training requires $6B(\tau N_{\text{small}} + (T-\tau)N_{\text{large}})$, which is significantly less if (I) $\tau$ is close to $T$ and (II) $N_{\text{small}} \ll N_{\text{large}}$. As a brief preview, we will develop techniques to push $\tau \approx 0.8T$ and to train zero/one-layer small models, hence accelerating by $\approx 5\times$ in Figure 1.

A long list of research has contributed to the development of progressive training, especially on initialization of large models, multi-stage training, training regime, and theory.

**Initialization from precedented small models.** (Chen et al., 2015; Wang et al., 2023b; Yao et al.) study the function-preserving initialization, such that the large model has the same loss and function as the small model at the moment of depth expansion. These works scale up the depth of convolution networks and BERT by $2\times$ and reduce the computation to $\approx 70\%$ computation. However, while function-preserving guarantees nice behavior during depth expansion, it does not guarantee fast convergence after the expansion. Alternatively, without function-preserving, (Chen et al., 2021) linearly combines two layers to initialize a new layer; (Gong et al., 2019; Yang et al., 2020; Du

et al., 2024) stacks the old layers multiple times; (Qin et al., 2021; Wang et al., 2023a) propose learning-based methods that require extra training. These methods empirically scale up the depth by $2 \sim 4\times$ and reduce the "grown v.s. target" computation to $\approx 55 \sim 70\%$ (e.g. (Wang et al., 2023a), Figure 1 in (Chen et al., 2021), Figure 6 in (Pan et al., 2024), and Figure 1 in (Du et al., 2024)). In contrast, we scale up the depth to $60\times$ and reduce the computation to $20\%$.

**Multi-stage training.** Most works in progressive training expand the small models once like (1.1). However, many works study multi-stage training and gradual stacking (Reddi et al., 2023). For instance, (Gong et al., 2019; Shen et al., 2022; Qin et al., 2021; Pan et al., 2024; Yao et al.) scale up the sizes of BERT for $3 \sim 4\times$ during training, optionally freezing some of the layers at some stages (Agarwal et al., 2024; Yang et al., 2020). We note that none of these multi-stage methods demonstrate the mixing behavior as we do.

**Training regime.** While most progressive training methods are tested on classification models like BERT (Devlin et al., 2019) and ViT (Dosovitskiy et al., 2020), some recent papers have evaluated on generative language models like GPT2 and reported $1.4 \sim 2\times$ speedup. As shown in Figure 1, our method enjoys $5\times$ speedup across different model sizes. We also note some papers that scale up MoE (not in terms of depth though) but the speedup seems transient as discussed in Section 7.

**Theory.** Theoretical analysis in progressive training is largely lacking, except (Agarwal et al., 2024) on strongly-convex and smooth loss. In contrast, we give a convergence theory of convex and Lipschitz continuous (non-smooth) loss and empirically validate its insights. Besides a convergence theory, we also study feature learning and hyperparameter transfer for progressive training.

## 1.1 RELATED WORK

In addition to previous works in progressive training, this work is closely related to convex optimization in Section 4, feature learning theory in Section 3.2, and learning rate schedules (especially warmup-stable-decay; WSD (Xing et al., 2018; Hägele et al., 2024)).

## 1.2 CONTRIBUTIONS

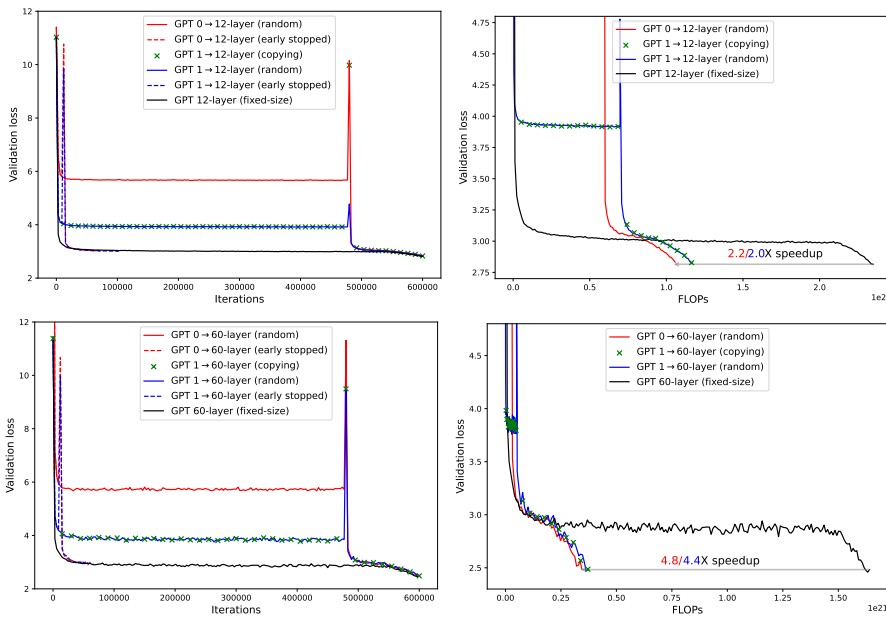

**Figure 1** Zero-layer (red, 39M or 0.15B) and one-layer (blue, 46M or 0.27B) progressive training can achieve significant speedup over fixed-sized training (black, 12-layer 124M or 60-layer 7B) on GPT2 pre-trained on OpenWebText under WSD schedule. The difference in final validation loss is $< 0.5\%$ for 124M runs and $< 0.2\%$ for 7B runs. The depth expansion takes place at 80% of iterations for full runs, and at 2% of iterations immediately after warmup for early stopped runs.

To our best knowledge, we are the first to advocate zero/one-layer depth expansion and to explore WSD schedule in progressive training. We summarize our main contributions here and provide extra insights on optimizer choice, optimizer states, model size, batch size, multi-stage expansion, and other topics in Section C.

1. We analyze the depth expansion as an initialization problem and ensure feature learning. This approach allows hyperparameter transfer (e.g. learning rate) throughout the progressive training, in contrast to extra hyperparameter tuning (Gu et al., 2020; Yano et al., 2025).

2. We reveal the important role of learning rate schedule, especially WSD schedule which theoretically and empirically improves the convergence.

3. We discover the mixing behaviors of progressive training, which supports the mixing time transfer and single-stage training, in contrast to multi-stage expansion (Gong et al., 2019; Yang et al., 2020; Qin et al., 2021; Shen et al., 2022; Pan et al., 2024).

4. We show that zero/one-layer progressive training has the best tradeoff between computational cost and loss, compared to other progressive or fixed-size training (see Figure 8). Specifically, zero/one-layer depth expansion is easy to implement and avoids the ordering problem (see Section 3.3 and Section A.3).

5. We theoretically analyze the convergence of progressive training under convex optimization to give insights on initialization, learning rate schedule, and projected gradient descent.

## 2 EXPERIMENT SETTINGS

We use the GPT2 (Radford et al., 2019) and ResNet (He et al., 2016) model families as testbeds[1]. For GPT2, we train on OpenWebText dataset (Gokaslan et al., 2019) with 1024 sequence length, following the nanoGPT codebase. For ResNet, we train on ImageNet dataset with $224 \times 224$ resolutions for 100 epochs.

Our main optimizer is Muon-NSGD, with 0.1 weight decay and without gradient clipping. The Muon-NSGD is adapted from the original Muon (Jordan et al., 2024) by (1) optimizing all 2D tensors with Muon and other tensors with normalized SGD (NSGD), and (2) using a single learning rate for Muon and NSGD. We use the cosine learning rate schedule and WSD schedule, that decay to 0 with 2% warm-up. Additional details are in Section B.

## 3 HOW TO EXPAND DEPTH?

### 3.1 DEPTH EXPANSION APPROACHES

We introduce multiple approaches to expand the depth of a residual neural network.

- **[copying]**: New layers are copied from the small model (Chang et al., 2018; Gong et al., 2019; Li et al., 2020).
- **[random]**: New layers are randomly initialized (Wang et al., 2017; Chen et al., 2021).
- **[zero]**: New layers are initialized as zeros. This approach kills the gradient flow and makes the new layers untrainable, hence invalidating the progressive training .
- **[copying_zero]**: New layers are copied from the small model, except some sub-layers are zero (Shen et al., 2022; Wang et al., 2023b; Tan et al., 2024; Wu et al., 2024; Du et al., 2024).

To test these approaches in a minimalist manner, we expand zero/one-layer versions of GPT2 and ResNet to multiple layers in Figure 2. We further experiment on LLAMA3 (Dubey et al., 2024),

---

[1]For GPT2, the models are configured as [Embedding, Hidden$\times N$, LM_head (with LayerNorm)], where 'Hidden' stands for the transformer layer and $N$ is number of layers, e.g. zero/one-layer GPT2 is $N = 0$ or 1. For ResNet, the models are configured by 4 stages, e.g. ResNet50 with [3,4,6,3] and ResNet101 with [3,4,23,3]. In each stage, the first layer has one shape, and each of the other layers has the same shape which is different to the first layer. Hence the zero-layer analogy corresponds to ResNet14 with [1,1,1,1] and the one-layer analogy corresponds to ResNet26 with [2,2,2,2].

Qwen3 (Yang et al., 2025), Mixtral (Jiang et al., 2024), and DeepseekV3 (Liu et al., 2024) to validate on modern architectures with 0.25B parameters. The experiments of copying_zero approach can be found in Section A, and more experiments on Mixture of Experts (MoE, (Fedus et al., 2022)) in Section C.

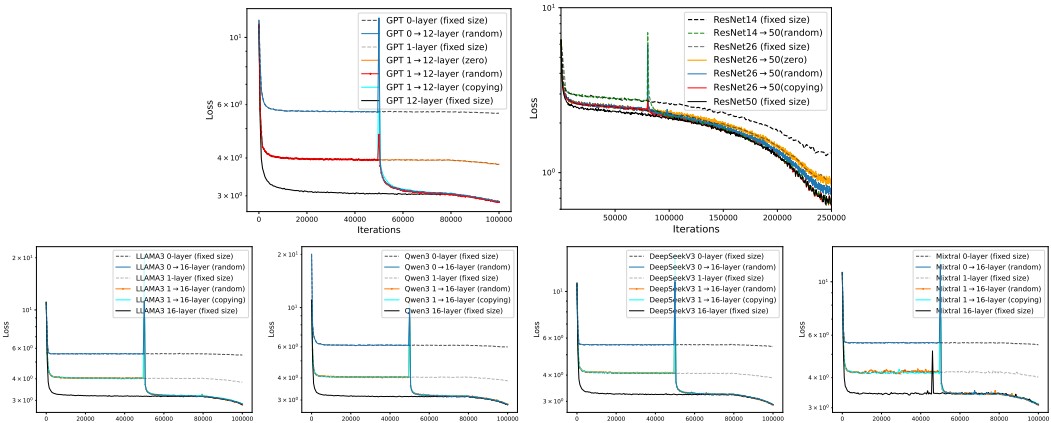

**Figure 2** Convergence of zero/one-layer progressive training and fixed-size training. Top left: GPT2 with depth expansion at 50k iterations. Top right: ResNet with depth expansion at 32-th epoch. Bottom left to right: LLAMA3 (dense), Qwen3 (dense), DeepseekV3 (MoE), Mixtral (MoE) with depth expansion at 50k iterations.

> **Takeaway 1:** For zero/one-layer progressive training, *random* and *copying* are empirically the best initializations of new layers.

## 3.2 FEATURE LEARNING AND HYPERPARAMETER TRANSFER

To ensure feature learning and keep the representations non-trivial and stable, each layer's activation needs to have consistent element sizes: denoting $l$-th layer's activation as $\mathbf{A}_l \in \mathbb{R}^{n_l}$, then $\|\mathbf{A}_l\|_2/\sqrt{n_l} \sim \|\mathbf{A}_{l+1}\|_2/\sqrt{n_{l+1}}$ (Mei et al., 2019; Yang & Hu, 2020; Chizat & Bach, 2018; Yang et al., 2022; 2023). For linear layers $\mathbf{A}_{l+1} = \mathbf{A}_l\mathbf{W}_l$, this translates to the spectral scaling condition by muP theory, i.e. the spectral norm $\|\mathbf{W}_l\|_* \sim \sqrt{n_{l+1}/n_l}$ for all layers.

Importantly, muP allows zero-shot hyperparameter transfer across model sizes, so that the optimal hyperparameters (e.g. learning rate) are the same for small and large models. In Figure 3, we illustrate this on the Muon-NSGD optimizer with muP-scaling learning rate. We highlight that hyperparameter transfer is particularly desirable in progressive training, where model sizes change significantly before and after the model expansion.

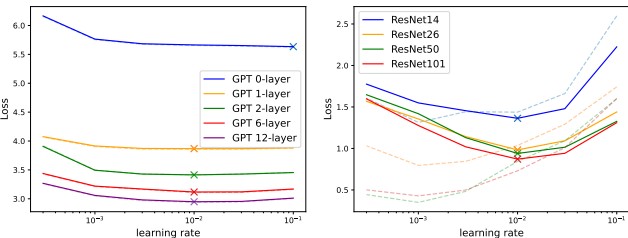

**Figure 3** Validation (solid) and training loss (dashed) at different learning rates of Muon-NSGD.

In fact, without the residual connection, only *copying* and *random* satisfy muP condition, but not *zero* or *copying_zero*, since any zero sub-layer will have $\|\mathbf{W}_l\|_* = 0 \not\sim \sqrt{n_{l+1}/n_l}$. Consequently, there is a conflict between muP condition and function-preserving (FPI) outside the residual path.

Here function-preserving means the large model has exactly the same loss as the small model (Chen et al., 2015; Shen et al., 2022; Wang et al., 2023b), hence no loss spikes.

**Table 1**  Summary of initialization approaches in progressive training.

|          | function-preserving | trainability | feature learning |
|----------|:-------------------:|:------------:|:----------------:|
| copying  | no                  | high         | yes              |
| random   | no                  | high         | yes              |
| zero     | yes                 | low          | no               |

> **Takeaway 2:**  For layers outside the residual path, zero initialization $\implies$ FPI $\implies$ not muP, which does not enjoy fast convergence and feature learning.

### 3.3  WHERE TO EXPAND?

For zero-layer expansion, only *random* initialization works; for one-layer expansion, *random* and *copying* both work; however, for multi-layer expansion, we must consider the ordering in depth expansion. We consider three variants of *copying* initialization: suppose we expand 3 to 6 layers,

- **[copying_last]**, copying only the last layer, e.g. $[1, 2, 3] \to [1, 2, 3, 3, 3, 3]$.
- **[copying_stack]**, copying and stacking all layers, e.g. $[1, 2, 3] \to [1, 2, 3, 1, 2, 3]$.
- **[copying_inter]**, copying and interpolating all layers, e.g. $[1, 2, 3] \to [1, 1, 2, 2, 3, 3]$.

We note that copying_inter is adopted by (Chang et al., 2018; Pan et al., 2024; Dong et al., 2020; Qin et al., 2022), as well as (Wang et al., 2023b) if some sub-layers are zeros; copying_stack is adopted by (Gong et al., 2019; Li et al., 2020; Fu et al., 2023), as well as (Shen et al., 2022; Du et al., 2024) if some sub-layers are zeros.

To test these variants, we experiment with deeper models such as ResNet50 and GPT 6-layer in Figure 4. We observe that copying all layers is consistently better than only copying one layer (copying_last), whereas copying_inter and copying_stack are almost indistinguishable.

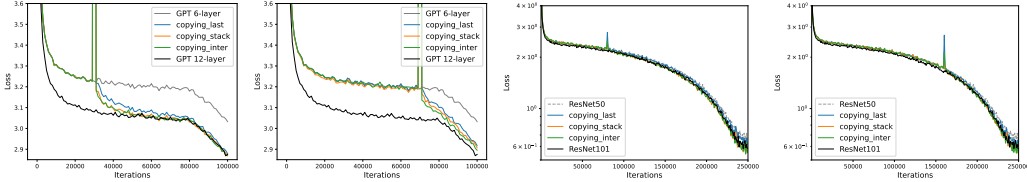

**Figure 4**  Convergence of multi-layer progressive training and fixed-size training. Left: ResNet with depth expansion at 32/64-th epoch. Right: GPT2 with depth expansion at 30/70k iterations.

> **Takeaway 3:**  *Copying_inter* and *copying_stack* are similarly performing for multi-layer depth expansion, but they are invalid for zero-layer depth expansion and equivalent for one-layer depth expansion (e.g. from $[1] \to [1, 1, 1, 1, 1, 1]$).

## 4  A CONVERGENCE THEORY FOR PROGRESSIVE TRAINING

We analyze the convergence of progressive training under convex and $G$-Lipschitz loss $L$. In fact, although deep learning is non-convex, its training dynamics is similar to convex optimization (Schaipp et al., 2025; Defazio & Mishchenko, 2023; Lee et al., 2019; Bu et al., 2021; Jacot et al., 2018; Allen-Zhu et al., 2019; Leclerc & Madry, 2020; Bu & Xu, 2024) and our analysis offers useful insights for the training recipe in Section 7.

We denote the small model before depth expansion as $\mathbf{w}_t$, the large model after depth expansion as $\mathbf{W}_t$, and the corresponding minima as $\mathbf{w}^*$ and $\mathbf{W}^*$. For any iteration trained with SGD, $\mathbf{w}_{t+1} =$

$\mathbf{w}_t - \eta_{t+1}\frac{\partial L}{\partial \mathbf{w}_t}$, the classical analysis gives

$$\|\mathbf{w}_{t+1} - \mathbf{w}^*\|^2 \leq \|\mathbf{w}_t - \mathbf{w}^*\|^2 - 2\eta_{t+1}(L(\mathbf{w}_t) - L(\mathbf{w}^*)) + \eta_{t+1}^2 G^2 \tag{4.1}$$

and equivalently, for the large model training with the same learning rate schedule,

$$\|\mathbf{W}_{t+1} - \mathbf{W}^*\|^2 \leq \|\mathbf{W}_t - \mathbf{W}^*\|^2 - 2\eta_{t+1}(L(\mathbf{W}_t) - L(\mathbf{W}^*)) + \eta_{t+1}^2 G^2 \tag{4.2}$$

Now for the progressive training with depth expansion at $t = \tau$, we use telescoping sum (4.1) from $t = 0 \rightarrow \tau - 1$ and (4.2) from $t = \tau \rightarrow T - 1$ to obtain

$$\|\mathbf{w}_\tau - \mathbf{w}^*\|^2 + \|\mathbf{W}_T - \mathbf{W}^*\|^2 \leq \|\mathbf{w}_0 - \mathbf{w}^*\|^2 + \|\mathbf{W}_\tau - \mathbf{W}^*\|^2 + \sum_{t=0}^{T-1} \eta_{t+1}^2 G^2$$

$$+ 2\sum_{t=0}^{\tau-1} \eta_{t+1}(L(\mathbf{w}^*) - L_t) + 2\sum_{t=\tau}^{T-1} \eta_{t+1}(L(\mathbf{W}^*) - L_t)$$

where $L_t = L(\mathbf{w}_t)$ for $t < \tau$, and $L_t = L(\mathbf{W}_t)$ otherwise.

Dividing by $2\sum_{t=0}^{T-1} \eta_{t+1}$ and re-arranging give

$$\frac{\sum_{t=0}^{T-1} \eta_{t+1}L_t}{\sum_{t=0}^{T-1} \eta_{t+1}} \leq \frac{\sum_{t=0}^{\tau-1} \eta_{t+1}L(\mathbf{w}^*) + \sum_{t=\tau}^{T-1} \eta_{t+1}L(\mathbf{W}^*)}{\sum_{t=0}^{T-1} \eta_{t+1}} + \frac{G^2 \sum_{t=0}^{T-1} \eta_{t+1}^2}{2\sum_{t=0}^{T-1} \eta_{t+1}}$$

$$+ \frac{\|\mathbf{w}_0 - \mathbf{w}^*\|^2 - \|\mathbf{W}_T - \mathbf{W}^*\|^2 + \|\mathbf{W}_\tau - \mathbf{W}^*\|^2 - \|\mathbf{w}_\tau - \mathbf{w}^*\|^2}{2\sum_{t=0}^{T-1} \eta_{t+1}}$$

On the left hand side, we apply the Jensen's inequality to get

$$L(\bar{\mathbf{W}}_T^{\text{progressive}}) \leq \frac{\sum_{t=0}^{T-1} \eta_{t+1}L_t}{\sum_{t=0}^{T-1} \eta_{t+1}} \text{ where } \bar{\mathbf{W}}_T^{\text{progressive}} = \frac{\sum_{t=0}^{\tau-1} \eta_{t+1}[\mathbf{w}_t, \mathbf{0}] + \sum_{t=\tau}^{T-1} \eta_{t+1}\mathbf{W}_t}{\sum_{t=0}^{T-1} \eta_{t+1}}$$

is the running average of iterates, and we have used $L([\mathbf{w}_t, \mathbf{0}]) = L(\mathbf{w}_t)$ for residual networks.

On the right hand side, we throw away $-\|\mathbf{W}_T - \mathbf{W}^*\|^2$ because it is small and negative. We obtain

$$L(\bar{\mathbf{W}}_T^{\text{progressive}}) \leq \frac{\sum_{t=0}^{\tau-1} \eta_{t+1}L(\mathbf{w}^*) + \sum_{t=\tau}^{T-1} \eta_{t+1}L(\mathbf{W}^*)}{\sum_{t=0}^{T-1} \eta_{t+1}} + \frac{G^2 \sum_{t=0}^{T-1} \eta_{t+1}^2}{2\sum_{t=0}^{T-1} \eta_{t+1}}$$

$$+ \frac{\|\mathbf{w}_0 - \mathbf{w}^*\|^2 + \|\mathbf{W}_\tau - \mathbf{W}^*\|^2 - \|\mathbf{w}_\tau - \mathbf{w}^*\|^2}{2\sum_{t=0}^{T-1} \eta_{t+1}} \tag{4.3}$$

We can easily recover the fixed-size large model training from scratch by setting $\tau = 0$:

$$L(\bar{\mathbf{W}}_T^{\text{fixed-size}}) \leq L(\mathbf{W}^*) + \frac{G^2 \sum_{t=0}^{T-1} \eta_{t+1}^2}{2\sum_{t=0}^{T-1} \eta_{t+1}} + \frac{\|\mathbf{W}_0 - \mathbf{W}^*\|^2}{2\sum_{t=0}^{T-1} \eta_{t+1}} \tag{4.4}$$

Subtracting the upper bounds in (4.3) from (4.4) , we would like the difference to be $\lessgtr 0$, and we write it as

$$\frac{\sum_{t=1}^{\tau} \eta_t(L(\mathbf{w}^*) - L(\mathbf{W}^*))}{\sum_{t=1}^{T} \eta_t} + \frac{\|\mathbf{w}_0 - \mathbf{w}^*\|^2 - \|\mathbf{W}_0 - \mathbf{W}^*\|^2 + \|\mathbf{W}_\tau - \mathbf{W}^*\|^2 - \|\mathbf{w}_\tau - \mathbf{w}^*\|^2}{2\sum_{t=1}^{T} \eta_t}$$

We view the large model as the concatenation of a small model and extra parameters $\mathbf{W}_t = [\mathbf{w}_t, \mathbf{x}_t]$ for $t = 0, \tau$, and simplify the analysis by assuming $\mathbf{W}^* = [\mathbf{w}^*, \mathbf{x}^*]$. We now obtain

$$\text{gap between upper bounds} = \frac{\sum_{t=1}^{\tau} \eta_t}{\sum_{t=1}^{T} \eta_t}(L(\mathbf{w}^*) - L(\mathbf{W}^*)) + \frac{\|\mathbf{x}_\tau - \mathbf{x}^*\|^2 - \|\mathbf{x}_0 - \mathbf{x}^*\|^2}{2\sum_{t=1}^{T} \eta_t}. \tag{4.5}$$

From the viewpoint of large model, we can mathematically view the progressive training as projected gradient descent (PGD) that masks deeper layers to zero, followed by an instant teleportation of $\mathbf{x}_\tau$ from zero to good initialization, then continued with SGD. In words, the effectiveness of progressive training comes from both optimizers (PGD and SGD) and teleportation of deeper layers.

Taking a closer look at (4.5), we can optimize this difference via the following factors.

- Initialization strategy of $\mathbf{x}_\tau$: given that $\mathbf{x}_0$ is randomly initialized, (1) if we randomly initialize new layers, then the second term is zero; (2) if we initialize better than random (e.g. copying), then the second term is negative and the difference is improved. This analysis is visualized in Figure 2.

- Learning rate schedule $\eta_t$: to minimize $\frac{\sum_{t=1}^{\tau} \eta_t}{\sum_{t=1}^{T} \eta_t}$, we prefer smaller $\eta_t$ for $t \leq \tau$ than for $t > \tau$, contrary to learning rate decay but consistent with WSD schedule, where $\eta_t$ remains constant during most iterations (see Figure 5).

To validate our insights on learning rate schedules, we experiment cosine and WSD schedules each with optimally tuned learning rate in Figure 5. We expand small models to large models at every 10% of total training horizon. For ResNet, the small model can still catch up with large model when $\tau \approx 0.8T$ under WSD schedule, but it fails to catch up around $\tau \geq 0.7T$ under cosine schedule; for GPT, the small model can catch up until $\tau \approx 0.8T$ under WSD schedule, but it fails around $\tau \geq 0.5T$ under cosine schedule.

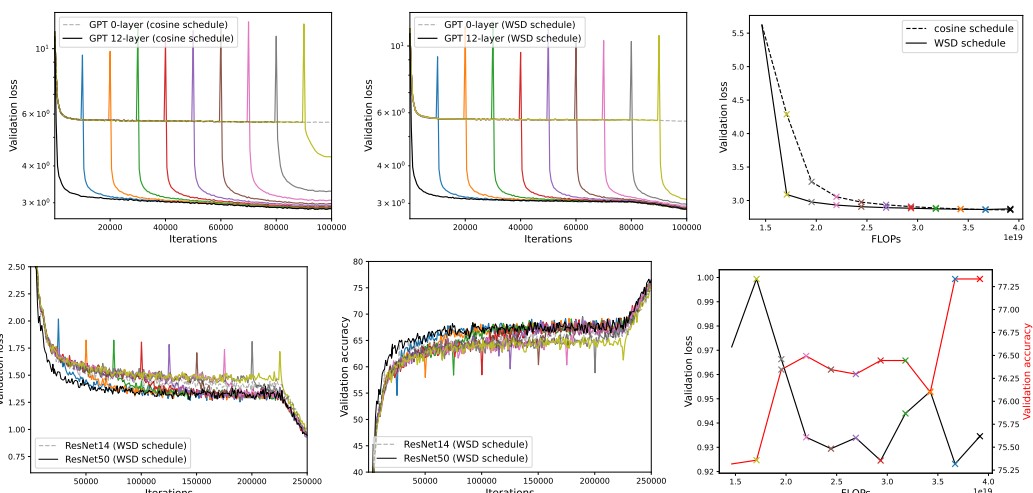

**Figure 5** Performance of zero-layer progressive training and fixed-size training, where WSD schedule significantly enhances the progressive training. See one-layer results in Section C.

> **Takeaway 4:** Progressive training is indeed "PGD + initialization of new layers + SGD", whose effectiveness relies on good initialization (e.g. random) and learning rate schedule (e.g. WSD).

## 5  WHEN TO EXPAND DEPTH?

To determine the timing of depth expansion, we need to understand the *mixing time*, which is the time until the loss of progressive training is close to the fixed-size large model training. To be specific, we define $t_{\mathrm{mix}}$ such that $L(\mathbf{W}_{\tau+t_{\mathrm{mix}}}^{\mathrm{fixed\text{-}size}}) \approx L(\mathbf{W}_{\tau+t_{\mathrm{mix}}}^{\mathrm{progressive}})$. Clearly, if the mixing time is short, then we can expand the models at later stage and save more compute.

### 5.1  PERSPECTIVES MATTER TO MIXING BEHAVIORS

We highlight that the mixing behaviors of progressive training (e.g. Figures 5, 9,13) have not been clearly observed in the literature, possibly due to the difference in perspectives of comparison.

In figures of (Wang et al., 2023a; Chen et al., 2021; Pan et al., 2024; Du et al., 2024), the comparison is between the grown model and the target model, while our comparison is based on the entire training (source and grown models). Such a perspective omits the computational cost of small models and the stated speedup must be discounted in our context. We re-plot Figure 5 (GPT under WSD schedule) from their perspective and no longer observe the mixing behaviors in Figure 6.

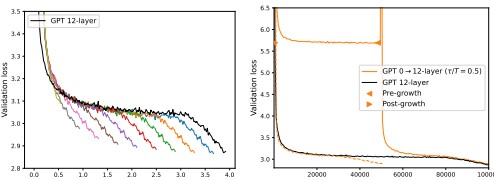
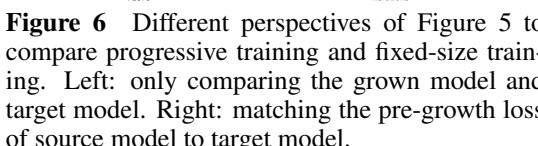
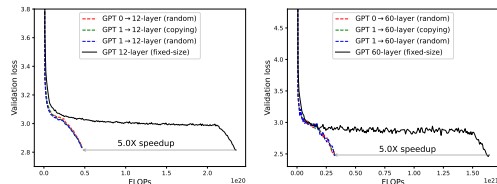

**Figure 6** Different perspectives of Figure 5 to compare progressive training and fixed-size training. Left: only comparing the grown model and target model. Right: matching the pre-growth loss of source model to target model.

**Figure 7** Comparing progressive training and fixed-size training via the grown model and target model of Figure 1. Left: 124M model. Right: 7B model.

Another perspective is to "overlay the loss curve for the grown model over the target model". (Shen et al., 2022) suggest that the convergence rate of grown model is "the same as the target model trained from scratch", and that the training dynamics is preserved. However, we claim that our method significantly improves the training dynamics instead of preserving it, as shown in Figure 6 by comparing the dashed orange curve and the solid black curve.

> **Takeaway 5:** The mixing behaviors of loss and training dynamics are the highlight of our depth expansion, and only observable via the comparison between the entire progressive training and the fixed-size training from scratch.

## 5.2 SENSITIVITY TO $\tau$ UNDER DIFFERENT SCHEDULES

Interestingly, in Figure 5, the mixing time $t_{\mathrm{mix}}(\tau)$ is highly sensitive to the timing of depth expansion $\tau$ for cosine schedule, but robust to $\tau$ for WSD schedule. For example, expanding GPT 1-layer at 10% horizon (blue curve) and expanding at 60% horizon (brown curve) both need $\approx 16B$ tokens or $30k$ iterations to mix with 12-layer training. However, expanding at 80% horizon (grey curve) cannot mix well as the learning rate has decayed. The same patterns hold for ResNet as well.

As a consequence, we determine the timing of depth expansion as total duration of constant learning rates minus mixing time in Figure 1. To be more precise, our WSD uses 2% warmup, 10% decay, and 528k iterations with constant learning rate. We subtract $\approx 40k$ iterations of mixing time from it (derived from Figure 5 or the early stopped run), and set the timing of depth expansion at $t = 480k$.

> **Takeaway 6:** During the stable phase of WSD schedule, the mixing time is almost unaffected by the timing of depth expansion. Hence we can transfer the mixing time at early iterations until the decaying phase (see Figure 1).

## 6 WHICH TO EXPAND DEPTH?

While we can expand the depth of any small model, we show the following through 150 runs (3 large model sizes, 5 small model sizes, 10 expansion times) in Figure 8.

> **Takeaway 7:** It is the most computationally efficient to (I) scale up from the zero/one-layer models and (II) scale up only once, i.e. use single-stage progressive training.

As we see in Figure 8, the zero/one-layer progressive training almost captures the loss-compute tradeoff from a Pareto-optimal viewpoint, especially in contrast with the progressive training from more than 2 layers. Additionally, the latest timing of expansion that still allows the progressive training to mix with the fixed-size training is not sensitive to small model sizes. In other words, expanding from 1-layer or from 6-layer at $\tau/T \approx 0.6$ is similarly effective, but the latter is much more computationally expensive.

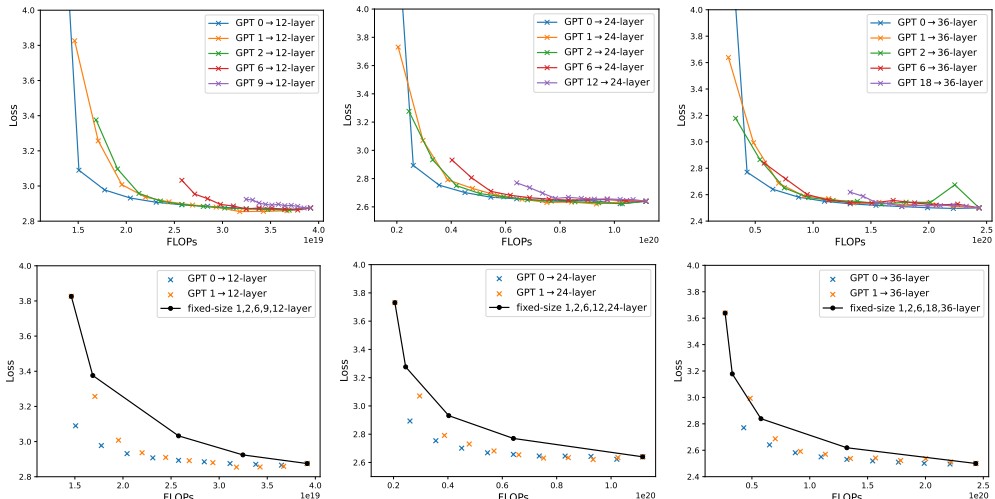

**Figure 8** Loss-compute tradeoff (validation loss v.s. FLOPs) of depth expansion from small models to $\{12, 24, 36\}$-layer GPT2 with $\{124M, 400M, 1B\}$ parameters.

Another insight from the loss-compute tradeoff is that, it suffices to use single-stage expansion, i.e. we do not need multi-stage expansion such as $0 \to 12 \to 24$ (if our target model is 24-layer). The reason lies in the mixing behaviors such that $0 \to 12 \to 24$ can be decomposed to two single-stage expansions $0 \to 12$ and $12 \to 24$. Therefore, the loss curve of $0 \to 12 \to 24$ will mix with those of $0 \to 12$ and $12 \to 24$, in the first stage and the second stage, respectively. As a result (see Section C.3), multi-stage training achieves the same loss with worse efficiency than single-stage training, in sharp contrast to (Gong et al., 2019; Yao et al.) where the mixing behaviors are not observed.

# 7 DEEP PROGRESSIVE TRAINING RECIPE

We summarize our progressive training recipe, leveraging the theoretical insights and empirical evidences in previous sections.

1. Train zero/one-layer model and then expand depth by random initialization[2].

2. Train models with Muon-NSGD (or other muP-scaled optimizers) and employ the same hyperparameters before and after depth expansion.

3. Train models with WSD learning rate schedule and expand depth during the stable phase.

4. The timing of depth expansion $\tau$ (or equivalently the mixing time $t_{\mathrm{mix}}$) can be determined by two small-scale runs: one fixed-size training and one progressive training ($\tau$ at the end of warmup), both early stopped when their losses mix.

We further validate our recipe with MoE (Wolfe, 2024; Xue et al., 2024) on OpenWebText dataset, and we observe the same patterns as dense models such as the mixing behaviors. We emphasize that our approach is different and orthogonal to existing works that upcycle MoE (He et al., 2024), which scale up a small dense model to a large MoE without increasing the depth, rather than a shallow MoE to a deep MoE. This upcycling approach has reported some negative results (Muennighoff et al.; Komatsuzaki et al.; Nakamura et al.; Liew et al.; Wei et al., 2024), because the grown MoE becomes worse than the MoE trained from scratch after a few hundred billion tokens.

---

[2]Alternatively, train one-layer model and expand by copying , e.g. $\mathbf{w} \to [\mathbf{w}, \mathbf{w}, \mathbf{w}]$.

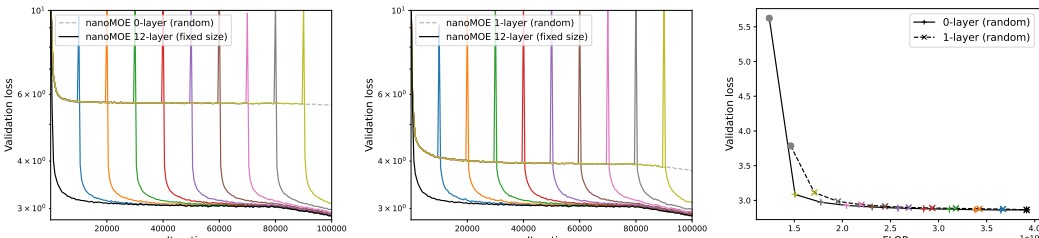

**Figure 9** Convergence of zero/one-layer progressive training and fixed-size training for MoE, with random initialization of new layers.

> **Takeaway 8:** Zero/one-layer progressive training is effective on various model architectures including ResNet, dense language models and MoE.

## 8 CONCLUSION

We show that zero/one-layer progressive training can significantly accelerate large-scale training, if it is equipped with good initialization method and learning rate schedule, and retain almost all the performance due to the mixing behaviors. This work demonstrates the power of theoretical insights into progressive training, drawing tools from feature learning and optimization theory. We expect future works to continue pushing the efficiency frontier, e.g. by scaling up both width and depth.

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

## A    DEPTH EXPANSION APPROACHES

### A.1    APPLICABILITY OF RANDOM, COPYING, AND ZERO INITIALIZATION

We summarize the depth expansion approaches with respect to the depth of source models.

**Table 2**    Applicability of depth expansion approaches. Merged cells indicate that multiple approaches are equivalent for a source model.

|  | zero-layer | one-layer | multi-layer |
|---|---|---|---|
| random | Yes | Yes | Yes |
| copying_inter |  |  | Yes |
| copying_stack | No | Yes | Yes |
| copying_last |  |  | Yes |
| zero | Yes | Yes | Yes |

We note that zero-layer or one-layer depth expansion significantly simplifies the copying approaches, as there is no ordering such as stacking or interpolation. In Appendix H of Du et al. (2024), the search space of ordering is enormous but necessary, since a proper ordering indeed improves the progressive training. This aligns with our results that copying all layers is better than copying only the last layer. However, it is debatable which of copying_inter and copying_stack is more advantageous, e.g. Pan et al. (2024) claims that copying_inter is more stable but Du et al. (2024) demonstrates that copying_stack converges better.

We highlight that such debate is completely avoided for zero-layer or one-layer progressive training.

### A.2    COPYING_ZERO INITIALIZATION

Completely zero initialization renders new layers not trainable, despite the depth expansion is function-preserving. It has been shown in the literature that copying with partially zero initialization has better trainability and is still function-preserving.

There are two known methods that copy all sub-layers except (1) the normalization sub-layers are initialized as zeros Shen et al. (2022), or (2) the last linear sub-layer is initialized as zero (Wang et al. (2023b); Tan et al. (2024); Wu et al. (2024) and Du et al. (2024) $G_{\text{zero}}$), or masked with zeros Yao et al.. These approaches are termed as copying_zeroN and copying_zeroL, which enforce zero output of a new layer and are function-preserving.

We experiment in the same setting as in Figure 2. Empirically, copying_zeroN has weak trainability, but copying_zeroL converges as fast as copying (without any zero sub-layers) and avoids any loss spike unlike copying.

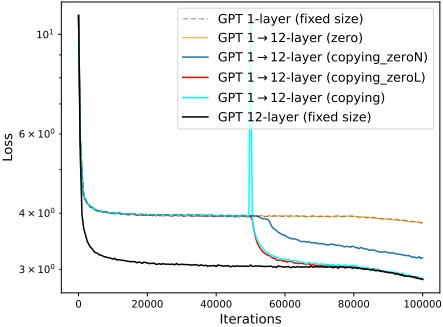

**Figure 10**    Convergence of one-layer progressive training and fixed-size training, with 2 different approaches of copying_zero initialization.

### A.3 ON THE PERFORMANCE AND ORDERING OF RANDOM INITIALIZATION

We observe that random initialization of new layers works well on GPT2 and MoE, but slightly less so on ResNet. We think the reason is the location of insertion: for GPT2 and MoE, the insertion is at the bottom, e.g. [1,2,3] to [1,2,3,R,R,R]; however, for ResNet with 4 stages, the insertion is intermittent since the model architecture is inhomogeneous, e.g. ResNet26 to ResNet50 is like [[1],[2],[3],[4]] to [[1,R],[2,R], [3,R,R,R],[4,R]].

On a related note, we analyze the ordering of random initialization. We train 6-layer or 12-layer GPT2 and expand the depth at $\tau = 0.1T$. We insert randomly initialized layers on top or bottom of old layers, i.e. $[1,2,3,4,5,6] \rightarrow [R,...,R,1,...,6]$ or $[1,2,3,4,5,6] \rightarrow [1,...,6,R,...,R]$.

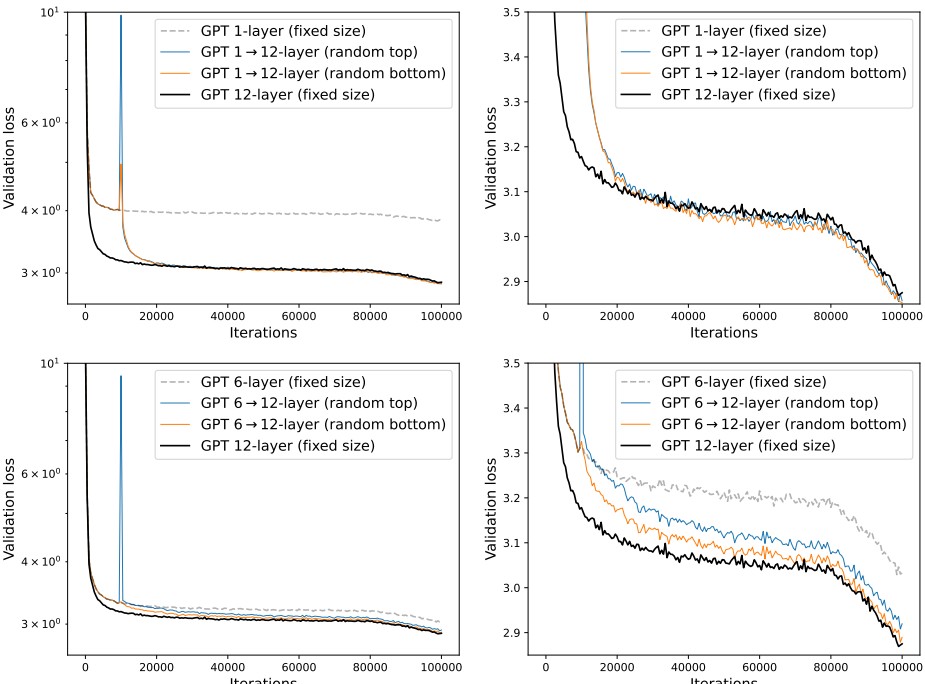

**Figure 11** Convergence of progressive training and fixed-size training with different insertion of random initialization (right plot is zoom-in of the left). Adding new layers at the bottom of old layers works best, with much smaller loss spikes.

We highlight that appending new layers at the bottom is empirically the best approach, and has much smaller loss spike than inserting at the top. However, such choice is completely avoided for zero-layer progressive training.

## B EXPERIMENT SETTINGS

Default batch size is 512 and decaying is 20% for WSD schedule, except for the long runs in Figure 1 where decaying is 10% and batch size=512 for 1B models and 64 for 7B models. For WSD schedule, the learning rate is 0.01 as shown to be optimal in Figure 3 (only here GPT2 are trained for 25k iterations); for cosine schedule, the learning rate is 0.05.

Regarding the optimizer, Muon-NSGD uses Muon Jordan et al. (2024) and NSGD as in Boreiko et al. (2025) in order to orthogonalize all tensors: denoting $\mathbf{W}$ as a layer's parameter, we apply

$$\text{Muon: } \mathbf{W}_{t+1} = (1 - \eta\lambda)\mathbf{W}_t - \eta \cdot \text{NS}(\mathbf{m}_t)$$
$$\text{NSGD: } \mathbf{W}_{t+1} = (1 - \eta\lambda)\mathbf{W}_t - \eta \cdot \mathbf{m}_t/\|\mathbf{m}_t\|_2$$

where NS is the Newton-Schulz matrix iteration, $\mathbf{m}_t$ is the momentum, and $\lambda$ is the weight decay.

For GPT2 models, we always keep n_embd/n_head=64. Different depth uses different n_head: full 12-layer uses 12 heads; full 24-layer uses 16 heads; full 36-layer uses 20 heads; full 60-layer uses 48 heads.

For MoE models, we use the same configurations as GPT2 (12-layer). Additionally, we use 8 experts, auxiliary loss Fedus et al. (2022), and router z loss Zoph et al. (2022).

For LLAMA3, hidden size=1024, intermediate size=2048, num attention heads=16, num key value heads=8, no weight tying; for Qwen3, hidden size=1024, intermediate size=2048, num attention heads=16, num key value heads=8, weight tying is used; for DeepseekV3, hidden size=512, intermediate size=1024, num attention heads=8, num key value heads=4, MLA and sparse attention is used; Mixtral, hidden size=512, intermediate size=1024, num attention heads=8, num key value heads=4.

## C  ADDITIONAL EXPERIMENTS

### C.1  COMPARING PROGRESSIVE TRAINING TO SHORTER FIXED-SIZE TRAINING

In Figure 1, we have observed that progressive training can achieve similar (though sometimes slightly higher) validation loss than fixed-size training under the same number of iterations. To make sure that progressive training has actual advantage instead of just moving along the loss-compute tradeoff, we launch another fixed-size training with shorter training horizon.

To be concrete, the depth expansion happens at $\tau = 0.8T$, meaning the grown model from progressive training is trained for 120k iterations. Our second fixed-size training runs for the same 120k iterations using the same learning rate schedule (60k in stable phase, 60k in decaying phase, no warmup, same peak learning rate).

It is clear that the progressive training does inherit from the small model training, despite the loss spike seems to suggest that all progresses before model expansion are lost. This is obvious by comparing the red and grey curves.

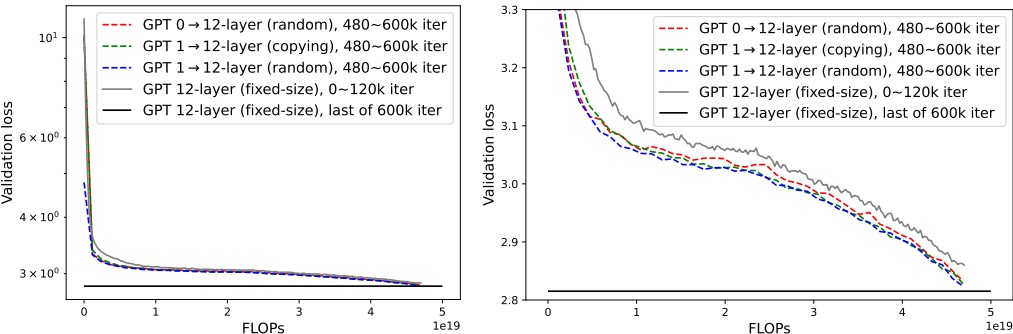

**Figure 12**  Comparing progressive training and fixed-size training via the grown model and target model (right plot is zoom-in of the left). Under the same compute budget, progressive training converges much faster than fixed-size training, despite having high loss after the depth expansion.

### C.2  MIXING BEHAVIORS ACROSS MODEL SIZES

In Figure 13, we consistently observe the mixing behaviors on hundreds of runs, from various small models to various large models. Specifically, the mixing time is empirically robust to model sizes.

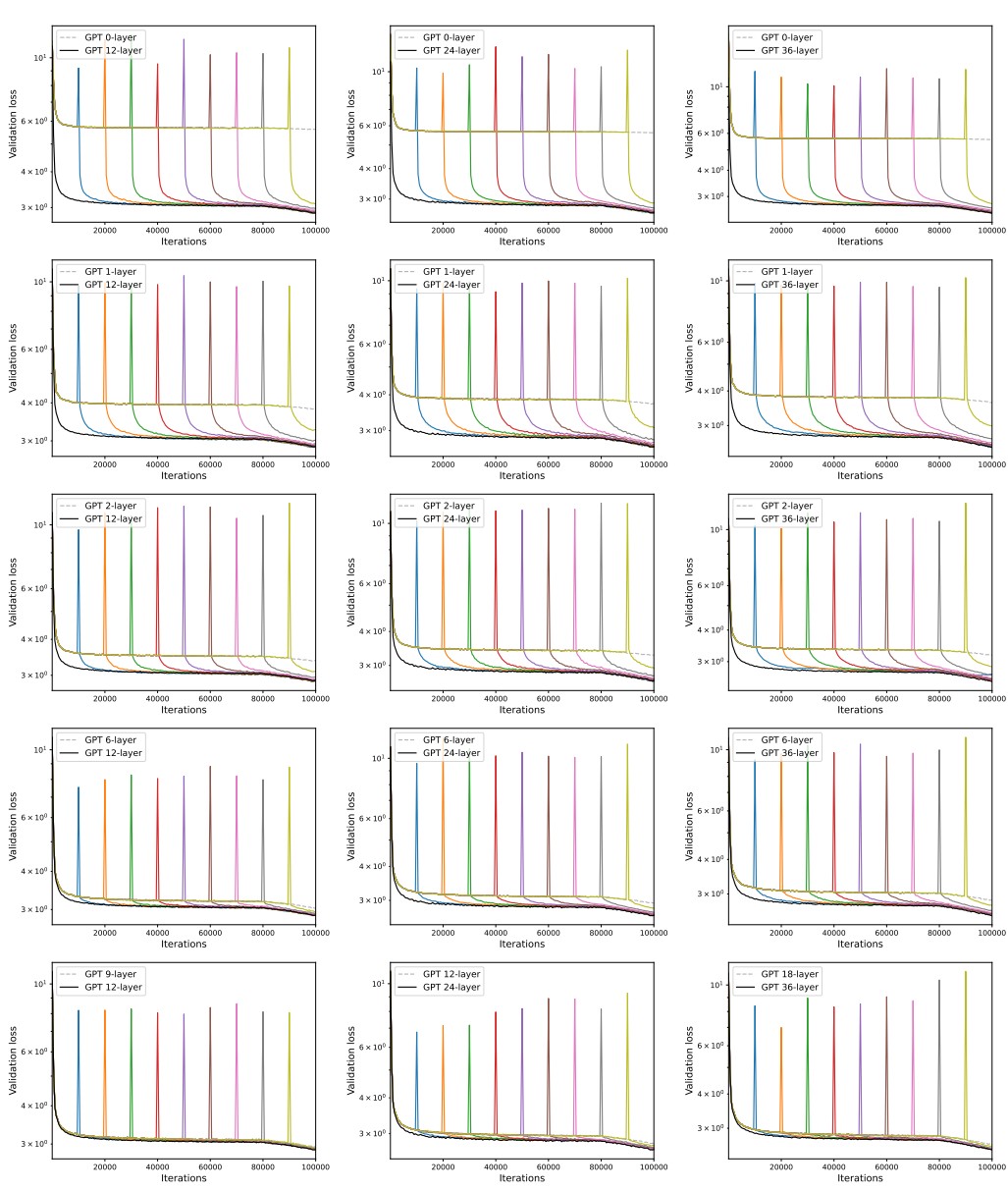

**Figure 13** Scaling up by depth expansion from $\{0, 1, 2, 6, 18\}$ layers to $\{12,24,36\}$ layers GPT2 with $\{124M, 400M, 1B\}$ parameters.

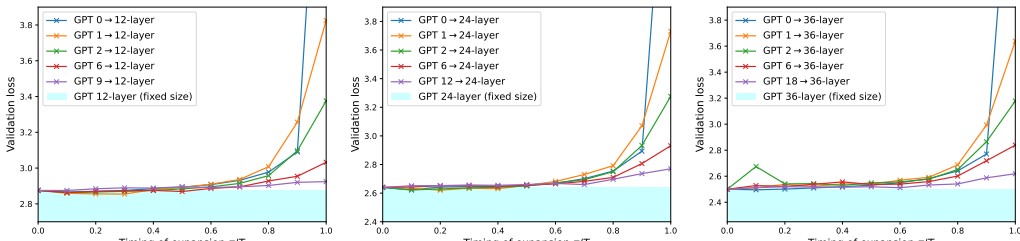

**Figure 14** Final loss of depth expansion at different timing, from $\{0, 1, 2, 6, 18\}$ layers to $\{12, 24, 36\}$ layers GPT2 with $\{124M, 400M, 1B\}$ parameters.

### C.3 SINGLE-STAGE PROGRESSIVE TRAINING IS SUFFICIENT

We experiment with single-stage progressive training (from 0-layer or 2-layer to 12-layer) and multi-stage training (from 0-layer to 2-layer then to 12-layer). As we expected, the mixing behaviors lead all runs to similar final losses, and multi-stage progressive training does not show improved efficiency: the multi-stage run has almost the same FLOPs as the 2-layer progressive training, worse than the 0-layer one. Additionally, multi-stage progressive training is hard to set up because of multiple timing of expansion.

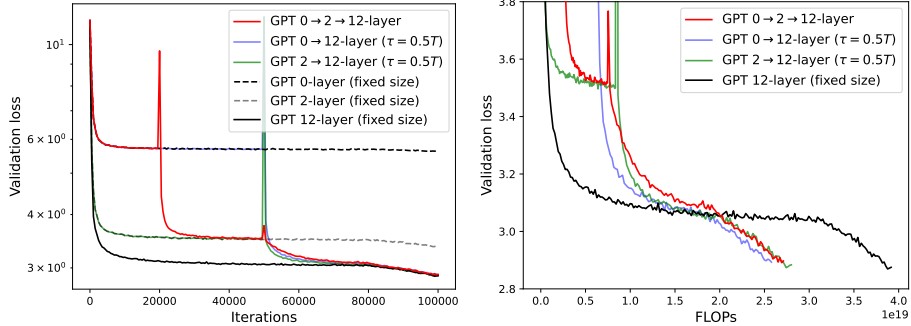

**Figure 15** Multi-stage progressive training does not show better efficiency or loss, due to the mixing behaviors.

## C.4 OPTIMIZER STATES

We conduct an ablation study to explore how to deal with the optimizer states (e.g. momentum and variance in AdamW or Muon-NSGD) during the expansion. Previous works have mixed results: Shen et al. (2022); Fu et al. (2023) show that copying the old layers' optimizer states to new layers can be helpful; Gong et al. (2019) resets the optimizer states of all layers.

We consider the following methods for optimizer states (OS): denoting embedding as E, hidden layers as H, and last layer as L,

- (inheriting OS) inheriting existing OS: $[E, H, L] \to [E, 0 \times 12, L]$

- (copying OS) inheriting existing OS and copying hidden layers' OS: $[E, H, L] \to [E, H \times 12, L]$

- (no OS) not inheriting any OS: $[E, L]$ or $[E, H, L] \to [0, 0 \times 12, 0]$

We observe that all methods seem to work in terms of final losses and mixing behaviors, although copying OS is less stable.

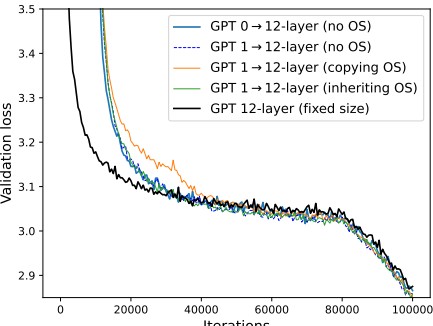

**Figure 16** Validation loss of depth expansion at $\tau = 0.1T$ with different ways to set optimizer states.

## C.5 CHOOSING OPTIMIZER AND LEARNING RATE SCHEDULE

In Figure 17, we train 100k iterations with two optimizers and two learning rate schedules. The same schedule is used before and after expansion, without changing the learning rate. We observe that Muon-NSGD with WSD schedule achieves best loss at all FLOPs (also at any timing of expansion $\tau/T$). This is consistent with our theory.

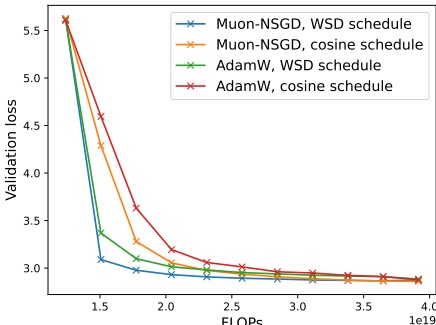

**Figure 17** Loss-compute tradeoff (validation loss v.s. FLOPs) of zero-layer depth expansion under different optimizers and learning rate schedules. The target model is 12-layer GPT2. For WSD schedule, AdamW uses 0.0005 learning rate and Muon-NSGD uses 0.01 learning rate. For cosine schedule, the learning rates are doubled.

## C.6 Mixing needs data, not iterations

Importantly, we observe that the mixing time is measured by data size, i.e. images or tokens processed, not by iterations. In Figure 18, we compare a progressive training with constant batch size to another one with $4\times$ batch size after the depth expansion ($\tau = 0.1T$). The final loss is similar although large-batch training takes much fewer iterations.

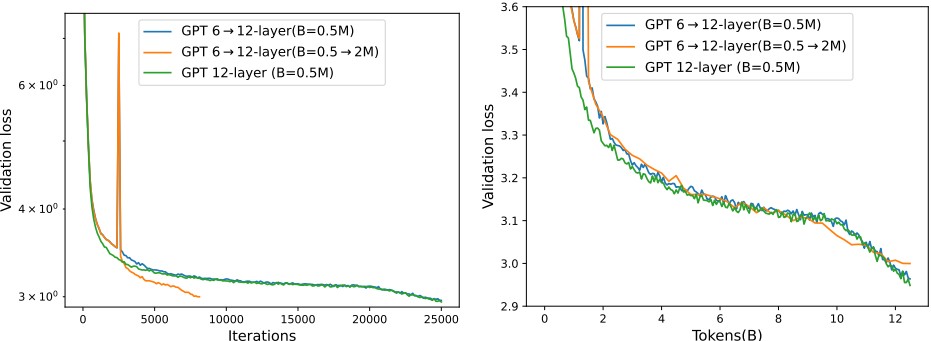

**Figure 18** Progressive training needs sufficient data to mix with fixed-size large model training, largely unaffected by batch size or iterations.

## C.7 One-layer model expansion figures

We present the one-layer model expansion results in correspondence to Figure 5 and Figure 6 here, due to space limit.

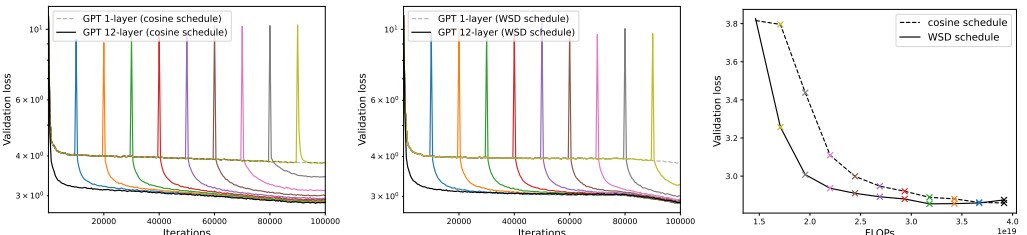

**Figure 19** Performance of one-layer progressive training and fixed-size training, where WSD schedule significantly enhances the progressive training.

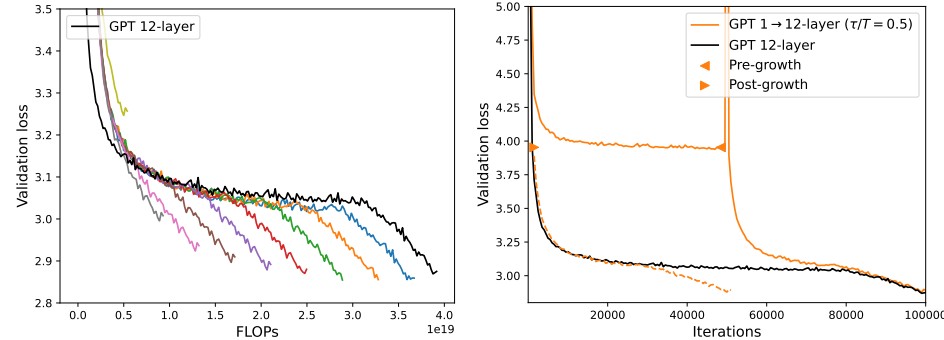

**Figure 20** Different perspectives to compare (one-layer) progressive training and fixed-size training. Left: only comparing the grown model and target model. Right: matching the pre-growth loss of source model to target model.

