# OpenReview forum: "Deep Progressive Training: scaling up depth capacity of zero-layer model"
_ICLR.cc/2026/Conference — Submitted to ICLR 2026_

### Official Review · Reviewer_2adh · 2025-10-24

**Soundness:** 2
**Presentation:** 2
**Contribution:** 2
**Rating:** 4
**Confidence:** 4

**Summary:**

This paper proposes a zero-layer progressive training framework to improve training efficiency for large-scale deep models. Instead of training a full-depth network from scratch, the method begins with a minimal (“zero-layer”) model and progressively expands its depth during training. The authors provide both theoretical analysis (based on convex optimization and feature learning theory) and empirical validation on GPT2, ResNet, and MoE architectures.

**Strengths:**

1. The paper introduces a novel and practical zero-layer progressive training paradigm that reduces computation cost by up to 5× with minimal accuracy loss.
2. The convergence analysis rigorously connects progressive training to convex optimization and projected gradient descent, adding rare theoretical depth.
3. Experiments across ResNet, GPT2, and MoE comprehensively validate the method under various initialization, expansion, and scheduling settings.

**Weaknesses:**

1.  The paper contains minor spelling mistakes such as “progrssive training” instead of progressive [L79] and “initialziation” instead of initialization [L214].
2. The paper does not include quantitative comparisons with related progressive or model expansion baselines (e.g., Net2Net, Gong et al. 2019, Yang et al. 2020, Wang et al. 2023a, Tan et al. 2024), which are discussed in Section 1.1–1.2 but not experimentally compared. This limits the ability to assess improvements over prior methods.
3. The experiments mainly evaluate GPT2 and ResNet models (Sections 2 and 7), but scalability to models larger than GPT2-7B or to other architectures such as multimodal or recurrent networks is not analyzed. Potential failure cases and computational constraints are not discussed.

**Questions:**

1. How does the proposed method perform when expanding both width and depth simultaneously?
2. Can the authors discuss potential instability in architectures without residual connections?
3. Is the WSD schedule critical, or could cosine warm restarts or adaptive schedulers achieve similar effects?

---

> ### Author Response · Authors · 2025-11-21
>
> Thank you for reading our paper and your feedback! Here are our point-to-point responses. If you are satisfied with our response, we would be grateful if you could raise your score.
>
> *---The paper contains minor spelling mistakes such as “progrssive training” instead of progressive [L79] and “initialziation” instead of initialization [L214].*
>
> Thank you for spotting the typos! We have corrected them.
>
> *---The paper does not include quantitative comparisons with related progressive or model expansion baselines (e.g., Net2Net, Gong et al. 2019, Yang et al. 2020, Wang et al. 2023a, Tan et al. 2024), which are discussed in Section 1.1–1.2 but not experimentally compared. This limits the ability to assess improvements over prior methods.*
>
> We would like to clarify that most of methods have been experimentally compared under different names (see Section 3.1, line 145 to line 153). To be concrete, Gong et al. is indeed the copying initialization; Net2Net, Wang et al., and Tan et al. is the copying\_zero initialization. Yang et al. is not explicitly experimented because it is similar to multi-stage expansion, which we compare in Appendix C.3. In particular, multiple prior methods are expanding both width and depth, hence these methods become the same method if we only expand the depth.
>
> *---The experiments mainly evaluate GPT2 and ResNet models (Sections 2 and 7), but scalability to models larger than GPT2-7B or to other architectures such as multimodal or recurrent networks is not analyzed. Potential failure cases and computational constraints are not discussed.*
>
> We hope the reviewer would understand that going beyond 7B models as well as extending to multimodal or recurrent networks is very expensive. For this work's scope, we choose to focus on the understanding of model expansion, e.g. the mixing behaviors and convergence analysis that are novel and not seen in the literature before. We agree these directions are important but will leave it to future work.
>
> We have not seen computational constraints on our progressive training, because the recipe is so simiple and compatible to regular fixed-size training! We would be happy to discuss any computational concerns the reviewer may have.
>
> *---How does the proposed method perform when expanding both width and depth simultaneously?*
>
> We have surveyed the literature and done some initial experiments: width expansion alone does not work as well as depth expansion alone (see Figure 3 of "Stacking Your Transformers: A Closer Look at Model Growth for Efficient LLM Pre-Training"). However, expanding width and depth simultaneously is empirically working in terms of mixing behaviors.
>
> *---Can the authors discuss potential instability in architectures without residual connections?*
>
> We haven't experimented on architectures without residual connections, as they are ubiquitous in modern neural networks. If we may make a conjecture, there should not be instability, as we have consistently observed (e.g. in Figure 1) that loss spike is not really a concern of instability in depth expansion.
>
> *---Is the WSD schedule critical, or could cosine warm restarts or adaptive schedulers achieve similar effects?*
>
> Our experiments have supported that WSD schedule is critical. In Equation (4.5), cosine warm restarts won't change the first fraction $\frac{\sum_{t=1}^{\tau}\eta_t}{\sum_{t=1}^{T}\eta_t}$, because it only affects $\eta_{t>\tau}$. We haven't tried adaptive schedulers since they are less commonly used in large-scale training.
>
> **New results:**
> We have new experiments on one-layer expansion with random initialization in Figure 1 (up to 7B param), Figure 7, Figure 9, as a strong alternative to copying initialization.
>
> We have new experiments on function-preserving expansion for the copying initialization in Appendix A.2. These emphasize that function-preserving may hurt trainability, extending our claim in Table 1. We also add Appendix C.3 (to demonstrate that single-stage progressive training is sufficient, in contrast to multiple previous papers where model depth is expanded multiple times during training), Appendix C.4 (to carefully validate how to deal with optimizer states during depth expansion), and Appendix C.7 (to additionally demonstrate the performance of one-layer depth expansion).

---

> > ### Comment · Reviewer_2adh · 2025-11-25
> >
> > Thank you for the detailed revision and responses. The authors’ replies have addressed part of my earlier concerns, and the newly added appendix materials do clarify several technical points.
> > However, some issues remain insufficiently explained. In particular, the experimental evidence is still limited to small GPT-2 and ResNet-scale models, with no validation on modern LLM architectures such as LLaMA, Qwen, Mixtral, or DeepSeek, making the generality of the proposed approach difficult to assess.
> > In addition, the paper still lacks a convincing justification for why zero/one-layer initialization is inherently optimal; this conclusion appears to be largely empirical without strong theoretical support. Given these unresolved questions, I will maintain my previous score.

---

> > > ### Author Response · Authors · 2025-11-27
> > >
> > > Thanks for your response! We kindly ask for some clarification to better address the reviewer's concern.
> > >
> > > 1. "the experimental evidence is still limited to small GPT-2 and ResNet-scale models...." Our experiments are not limited to *small* GPT-2 scale. Our Figure 1 has 7 billion parameter GPT-2. Could you confirm if 7B param is not large enough, and if so, what scale is desired?
> > >
> > > 2. The main concern "with no validation on modern LLM architectures such as LLaMA, Qwen, Mixtral, or DeepSeek.." is more about the architecture, not the scale. We are currently running LLaMA and Qwen experiments and will update soon (if we have time, we will run Mixtral and DeepSeek as well, though we do have MoE experiments that give some insights to Mixtral). Please let us know if these new experiments would be sufficient?
> > >
> > > 3. "the paper still lacks a convincing justification for why zero/one-layer initialization is inherently optimal". Did you mean why zero/one-layer initialization is optimal when compared to multi-layer initialization, or why random/copying init for zero/one-layer is optimal when compared to copying_zero or zero init?

---

### Official Review · Reviewer_v2fW · 2025-10-27

**Soundness:** 3
**Presentation:** 2
**Contribution:** 3
**Rating:** 6
**Confidence:** 2

**Summary:**

This paper introduces a progressive training strategy called Deep Progressive Training (DPT), which dynamically expands model depth during training to improve efficiency in large-scale model training. The authors analyze the depth expansion process from an optimization-theoretic perspective, offering insights into initialization strategies, hyperparameter transfer, and the timing of expansion. Experiments on GPT-2 and ResNet demonstrate that the proposed method achieves substantial improvements in training efficiency while maintaining competitive performance.

Key Reasons:
1. The paper presents a novel and practical direction in progressive training, particularly with the proposed zero-layer depth expansion strategy.
2. Combining theoretical insights with thorough experimental validation, the work provides meaningful inspiration and practical value for efficient large-scale training.


Supporting Arguments

The proposed zero-layer progressive training redefines conventional notions of “small models” in progressive training and offers a fresh conceptual perspective. The experimental design is rigorous, with validations across multiple models, scales, and stages, enhancing confidence in the approach. A simple and practical training recipe is also provided, which offers clear guidance for practitioners. However, the method has not yet been tested on popular large-scale LLM architectures such as LLaMA or Qwen, which would strengthen its impact.

**Strengths:**

1. The paper provides convergence analysis grounded in convex optimization theory, offering an interpretable framework for understanding progressive depth expansion.
Extensive experiments on GPT-2, ResNet, and MoE models demonstrate the generality and effectiveness of the approach across diverse architectures.
2. The proposed method achieves comparable or better performance with significantly reduced computational costs through a single-stage depth expansion strategy.

**Weaknesses:**

1. The convergence analysis is based on convex loss assumptions, which may limit its applicability to realistic non-convex deep learning settings.
2. The paper lacks direct comparisons with related efficient training methods such as knowledge distillation, model compression, or MoE-based strategies, which limits the comprehensiveness of the evaluation.
3. The ablation studies are limited — for example, the definition and impact of “mixing time,” and the stability of different initialization strategies at larger scales are not thoroughly explored.
4. The proposed method has not been evaluated on more recent large-scale architectures (e.g., LLaMA, Qwen), which would further validate its generality.

**Questions:**

1. Can zero-layer progressive training maintain the reported 5× acceleration at larger scales (e.g., >10B parameters) or on other architectures such as LLaMA or Qwen?
2. Have you considered integrating your approach with other efficient training techniques such as knowledge distillation, model compression, or MoE? Are there potential synergies or conflicts?
3. Since your theoretical analysis assumes convex loss functions, do you plan to extend the analysis to non-convex settings or derive tighter convergence bounds?
4. Has the method been evaluated on other modalities or task domains (e.g., multimodal learning)? If so, do you have preliminary evidence of its effectiveness?

---

> ### Author Response · Authors · 2025-11-21
>
> Thank you for reading our paper and your positive score! Here
> are our point-to-point responses. Please let us know if you want to extend the discussion.
>
> *---The convergence analysis is based on convex loss assumptions, which may limit its applicability to realistic non-convex deep learning settings.*
>
> We agree that deep learning is not convex. Interestingly, deep learning convergence can be convex-like to some degrees that still provide valuable insights. These insights are discussed in multiple references in the first paragraph of Section 4, especially "The Surprising Agreement Between Convex Optimization Theory and Learning-Rate Scheduling for Large Model Training" where LLAMA model trained by AdamW has similar loss shape to convex model trained with SGD.
>
> *---The paper lacks direct comparisons with related efficient training methods such as knowledge distillation, model compression, or MoE-based strategies, which limits the comprehensiveness of the evaluation.*
>
> Can the reviewer please explain a bit why these training methods are related? From our perspective, the progressive training is to train large models efficiently by starting from small models; knowledge distillation and model compression are doing the opposite: the end goals are small models and the starting points are large models.
>
> *---The ablation studies are limited — for example, the definition and impact of “mixing time,” and the stability of different initialization strategies at larger scales are not thoroughly explored. The proposed method has not been evaluated on more recent large-scale architectures (e.g., LLaMA, Qwen), which would further validate its generality.*
>
> We kindly remind the reviewer that "mixing time" is defined in the first paragraph of Section 5.
>
> We hope the reviewer would understand that going beyond 7B models is very expensive, and that our current experiments in different settings have not seen instability, except a single loss spike which only happens once at depth expansion and does not hurt long-term convergence. We view larger scale experiments on more architectures as a future direction, which will require much more work beyond the scope of this paper.
>
> *---Can zero-layer progressive training maintain the reported 5× acceleration at larger scales (e.g., >10B parameters) or on other architectures such as LLaMA or Qwen?*
>
> As discussed above, we will view this as future work due to limited compute resources.
>
> *---Have you considered integrating your approach with other efficient training techniques such as knowledge distillation, model compression, or MoE? Are there potential synergies or conflicts?*
>
> We would love to learn more from the reviewer, because our current understanding is that knowledge distillation and model compression are opposite to progressive training.
>
> *---Since your theoretical analysis assumes convex loss functions, do you plan to extend the analysis to non-convex settings or derive tighter convergence bounds?*
>
> Yes! We are working on tighter convergence bounds based on last iterate loss value, instead of the average iterate loss value, still under convex assumption. On the other hand, to move onto non-convex loss, we may need Lipschitz smoothness instead of Lipschitz continuity, though it is debatable which regime (L-smooth and non-convex, or L-continuous and convex) is closer to deep learning.
>
> *---Has the method been evaluated on other modalities or task domains (e.g., multimodal learning)? If so, do you have preliminary evidence of its effectiveness?*
>
> We have some early evidence that our method works on multimodal finetuning. We will continue to consolidate this in future work.
>
> **New results:**
> We have new experiments on one-layer expansion with random initialization in Figure 1 (up to 7B param), Figure 7, Figure 9, as a strong alternative to copying initialization.
>
> We have new experiments on function-preserving expansion for the copying initialization in Appendix A.2. These emphasize that function-preserving may hurt trainability, extending our claim in Table 1. We also add Appendix C.3 (to demonstrate that single-stage progressive training is sufficient, in contrast to multiple previous papers where model depth is expanded multiple times during training), Appendix C.4 (to carefully validate how to deal with optimizer states during depth expansion), and Appendix C.7 (to additionally demonstrate the performance of one-layer depth expansion).

---

### Official Review · Reviewer_Q6T9 · 2025-10-27

**Soundness:** 2
**Presentation:** 3
**Contribution:** 2
**Rating:** 4
**Confidence:** 4

**Summary:**

The paper studies deep progressive training in neural network pretraining. Extensive experiments including ResNets, transformers, and MoEs show the how, when, and which to perform depth expansion. Proposed configuration, which expands from zero-layers, provides 80% computational savings. The authors also provide theoretical justification for the insights with convex and Lipschitz continuous conditions.

**Strengths:**

1. The experiments are extensive, including different architectures, different learning rate schedules, and different expansion methods, which should provide great insights for researchers in the field. Authors also provide recommendations in Section 7 that are easy to implement.

**Weaknesses:**

1. The main concern is that there is possibly a derivation flaw in Section 4. More specifically, the authors try to obtain the difference of two losses by substracting two upper bounds. However, from 4.3,4.4 to 4.5, we cannot substract inequalities in opposite directions to get a valid bound on the difference. Hence, the theoretical justification as presented is incorrect and undermines the paper's theoretical contributions. (It is possible that I missed some intermediate steps, authors, please correct me if I am wrong.) Moreover, the condition of the derivation is too strong with convex and Lipschitz settings.
2. There are many typos and formatting issues in the paper, see questions. Zero-layer is also not defined in the paper.

**Questions:**

1. Figure 2 title: Left and right subfigures are mislabeled.
2. There are some citing format issues where \citet is used rather than \citep.
3. See weaknesses 1.

---

> ### Author Response · Authors · 2025-11-21
>
> Thank you for reading our paper and the feedback! Here are our point-to-point responses. If you are satisfied with our response, we would be grateful if you could raise your score.
>
> *---The main concern is that there is possibly a derivation flaw in Section 4. More specifically, the authors try to obtain the difference of two losses by substracting two upper bounds. However, from 4.3,4.4 to 4.5, we cannot substract inequalities in opposite directions to get a valid bound on the difference. Hence, the theoretical justification as presented is incorrect and undermines the paper's theoretical contributions. (It is possible that I missed some intermediate steps, authors, please correct me if I am wrong.)*
>
> We agree with the reviewer that one cannot subtract inequalities in opposite directions and we have corrected (4.5), by claiming that the gap between upper bounds can be small, not the gap between actual losses. Therefore, the worst-case losses (as indicated by the upper bounds) of progressive training and fixed-size training can be similar.
>
> We hope the reviewer would appreciate our theoretical analysis, which is novel and unique (to our best knowledge) in the field of progressive training, which has been historically empirical, given that we have corrected the flaws.
>
> *---Moreover, the condition of the derivation is too strong with convex and Lipschitz settings.*
>
> We agree that deep learning is not convex nor Lipschitz. Interestingly, deep learning convergence can be convex-like to some degrees that still provide valuable insights. These insights are discussed in multiple references in the first paragraph of Section 4, especially "The Surprising Agreement Between Convex Optimization Theory and Learning-Rate Scheduling for Large Model Training" where LLAMA model trained by AdamW has similar loss shape to convex model trained with SGD.
>
> *---There are many typos and formatting issues in the paper, see questions. Zero-layer is also not defined in the paper...Figure 2 title: Left and right subfigures are mislabeled.*
>
> Thank you for pointint these out. We have corrected them. We have now explained zero-layer (and one-layer) model architecture in Footnote 1 of page 3.
>
> *---There are some citing format issues where citet is used rather than citep.*
>
> Thank you. We have corrected these.
>
> **New results:**
> We have new experiments on one-layer expansion with random initialization in Figure 1 (up to 7B param), Figure 7, Figure 9, as a strong alternative to copying initialization.
>
> We have new experiments on function-preserving expansion for the copying initialization in Appendix A.2. These emphasize that function-preserving may hurt trainability, extending our claim in Table 1. We also add Appendix C.3 (to demonstrate that single-stage progressive training is sufficient, in contrast to multiple previous papers where model depth is expanded multiple times during training), Appendix C.4 (to carefully validate how to deal with optimizer states during depth expansion), and Appendix C.7 (to additionally demonstrate the performance of one-layer depth expansion).

---

> > ### Comment · Reviewer_Q6T9 · 2025-11-24
> >
> > My main concern is still about the theoretical contribution of the work. The revised manuscript still does not provide a meaningful theoretical justification. Showing that two upper bounds are close does not imply anything about the actual loss difference or the behavior difference of progressive vs. fixed-size training. Without a valid inequality relating the real losses, the theoretical section becomes mainly descriptive.

---

> > > ### Author Response · Authors · 2025-11-27
> > >
> > > Thank you for the response! We have two ways of presenting the theoretical insights. If the reviewer agrees, we will put both pieces of discussion in the paper.
> > >
> > > Firstly, we can give the rigorous bound between progressive training loss and fixed-size loss, because each loss is between 0 and some bound,
> > >
> > > $$\textup{negative of Eq.(4.4)}\leq L_{T}^{progressive}-L_{T}^{fixed-size}\leq \textup{Eq.(4.3)}$$
> > >
> > > We can make this simplified by substituting specific learning rate schedule, e.g. constant learning rate
> > >
> > > $$ -L(W^\*)-\frac{\eta G^2}{2}+O(\frac{1}{T})\leq L_{T}^{progressive}-L_{T}^{fixed-size}\leq L(W^\*)+\frac{\eta G^2}{2}+O(\frac{1}{T})$$
> > >
> > > Secondly, we can assume the bounds are tight, hence we remove the inequalities in the last step and "two upper bounds are close" means two losses are close, which is the case in practice. This is not rigorous but does explain the mixing behaviors that are the highlight of this work.
> > >
> > > We respectfully hope the reviewer can also take into account the broader contributions of our work. The manuscript includes substantial empirical and methodological advances, and the theory section was the only weakness identified. We believe that a single limitation should not outweigh the overall strengths of the paper, which we feel warrant an acceptance.

---

> > > > ### Comment · Reviewer_Q6T9 · 2025-11-28
> > > >
> > > > I would like to thank the authors for the additional clarification regarding the theoretical part of the work.
> > > >
> > > > **1. Regarding the “rigorous bound” in the rebuttal:**
> > > >
> > > > My understanding is that, although the bound is mathematically correct, it is not informative in my mind. It does not (1) show that the difference between the two losses converges at any rate, nor (2) establish which loss is larger, because the bounded interval spans both negative and positive values. In other words, the bound mainly shows that the difference between two bounded losses is itself bounded—this is true for any pair of bounded losses and thus does not provide meaningful insights into the progressive training research problem.
> > > >
> > > > **2. Regarding the assumption that the upper bounds are “tight”:**
> > > >
> > > > In fact, the authors do not provide an argument that the bounds are tight. Could the authors provide further discussion on why these bounds are tight?
> > > >
> > > > **3. Regarding the rating:**
> > > >
> > > > I do appreciate the empirical contributions of the paper, which provide interesting evidence and insights for the progressive training research community. However, the authors indicate that the convergence theory is an important part of the work, and indeed  the entire Section 4 is used for it. Given that the theoretical derivation contains a fundamental issue, I do not feel comfortable rating the paper above the acceptance threshold. Nevertheless, as reflected in my rating, I would not object if the paper is accepted based on its empirical contributions.

---

> > > > > ### Author Response · Authors · 2025-12-03
> > > > >
> > > > > Thank you for the extended discussion (while this conversation is impossible to continue due to recent leakage incident, we hope to provide more discussion here). We have some evidence that this bound is empirically tight, but this is for a separate paper and, due to anonymity policy, we will only be able to share it after this review cycle.
> > > > >
> > > > > Without relying on that, I can make a bound more informative and mathematically correct: note each loss is between $L_*$ and some bound (previously I was using between 0 and some bound!!),
> > > > >
> > > > > $$L_*-\textup{Eq.(4.4)}\leq L_{T}^{progressive}-L_{T}^{fixed-size}\leq \textup{Eq.(4.3)}-L_*$$
> > > > >
> > > > > We can make this simplified by substituting specific learning rate schedule, e.g. constant learning rate
> > > > >
> > > > > $$ -\frac{\eta G^2}{2}+O(\frac{1}{T})\leq L_{T}^{progressive}-L_{T}^{fixed-size}\leq \frac{\eta G^2}{2}+O(\frac{1}{T})$$
> > > > >
> > > > > This is not vanishing but could be very small for small lr, especially if $\eta$ scales with T.

---

### Author Response · Authors · 2025-12-03
**Summary of rebuttal and new results**

Dear AC and reviewers,

We appreciate your efforts in handling our paper. We summarize our responses that address each weakness raised by reviewers (questions are answered in original responses as well). We also highlight some new experiments that further strengthen this paper.

1. Reviewer Q6T9 raises only one main weaknesses "a derivation flaw in Section 4. More specifically, the authors try to obtain the difference of two losses by substracting two upper bounds. ... Moreover, the condition of the derivation is too strong with convex and Lipschitz settings."

We clarify that **deep learning convergence can be convex-like to some degrees that still provide valuable insights**. These insights are discussed in multiple references in the first paragraph of Section 4, especially "The Surprising Agreement Between Convex Optimization Theory and Learning-Rate Scheduling for Large Model Training" where LLAMA model trained by AdamW has similar loss shape to convex model trained with SGD.

**Our current derivation is insightful in the sense it explains the mixing behavior**, which is the key novelty in this work. However, to maximize it informativeness, we need an additional assumption that the upper bounds in (4.3) and (4.4) are somewhat tight, which is hard to justify. To work around it , **we give a new bound that is mathematically correct and informative** (see https://openreview.net/forum?id=G6scrBvQCL&noteId=w1bJ02AtlN) but we could not confirm which derivation is more appreciated due to the unexpected termination of author-reviewer discussion period.

2. Reviewer v2fW
 * Weakness 2: The convergence analysis is based on convex loss assumptions, which may limit its applicability to realistic non-convex deep learning settings.

Same as part of Reviewer Q6T9's concern, which is addressed by **deep learning convergence can be convex-like to some degrees that still provide valuable insights**.

* Weakness 3: "The paper lacks direct comparisons with related efficient training methods such as knowledge distillation, model compression, or MoE-based strategies, which limits the comprehensiveness of the evaluation."

**We believe this weakness is irrelevant as these methods are opposite to what we are doing**. The progressive training is to train large models efficiently by starting from small models; knowledge distillation and model compression start from large models and end up with small models.

* Weakness 4: "The ablation studies are limited — for example, the definition and impact of “mixing time,” and the stability of different initialization strategies at larger scales are not thoroughly explored."

We kindly remind the reviewer that **"mixing time" is already defined in the first paragraph of Section 5**. We also believe **going beyond 7B models is too expensive and not necessary**.

3. Reviewer 2adh

* Weakness 5: "The paper does not include quantitative comparisons with related progressive or model expansion baselines... which are discussed in Section 1.1–1.2 but not experimentally compared. This limits the ability to assess improvements over prior methods."

**This weakness is incorrect because these prior methods have been carefully compared in our paper**. These methods have been experimentally compared under different names (see Section 3.1, line 145 to line 153). To be concrete, Gong et al. is indeed the copying initialization; Net2Net, Wang et al., and Tan et al. is the copying_zero initialization. In particular, multiple prior methods are expanding both width and depth, hence these methods become the same method if we only expand the depth.

4. Shared weakness 6 of Reviewer 2adh and Reviewer v2fW

Both reviewers would like to see more model architectures. In addition to our GPT and MoE experiments, **we have added requested experiments on Qwen, LLAMA3, Deepseek V3, Mixtral in Figure 2**, which cover sparse attention, multi-head latent attention (MLA), RoPE, Grouped Query Attention (GQA), etc. We consistently observe 0/1-layer depth expansion to mix with fixed-size training.

5. **New results**

We have **new experiments on one-layer expansion with random initialization** in Figure 1 (up to 7B param), Figure 7, Figure 9, as a strong alternative to copying initialization. We also **added Appendix A.2, C.3, C.4, C.7** for various experiments (see details in last paragraph of https://openreview.net/forum?id=G6scrBvQCL&noteId=KJ7x1y8lLp)

*To summarize, out of all six weaknesses, weakness 3, 4 and 5 are invalid; others are properly addressed with new theoretical derivation or new experiments. We hope the AC can appreciate the novelty and impact of this work, but more importantly, hope everyone gets through this difficult submission cycle.*

---

### Meta-Review · Area_Chair_qbu2 · 2025-12-29

**Summary:**

The reviewers broadly agree that the paper explores an interesting and practically relevant direction—deep progressive training with zero- or one-layer depth expansion—and is supported by extensive empirical results across several architectures and training configurations. However, the primary concern driving the recommendation is the paper’s theoretical component, which is positioned as a key contribution but is widely viewed as flawed or insufficiently informative. In particular, the derivation in Section 4 relies on convex and Lipschitz assumptions that are difficult to justify for deep learning, and even after revisions, the analysis does not establish a meaningful or actionable relationship between progressive and fixed-size training losses. Additional concerns include the limited evaluation on truly modern LLM architectures and the lack of a convincing theoretical explanation for why zero/one-layer initialization should be optimal, leaving several core claims largely empirical.

**Reviewer Concerns:**

The rebuttal and revisions addressed a number of surface-level issues, including typos, presentation problems, clearer definitions of zero-layer models, and the addition of further ablations and appendix experiments, which improve clarity and completeness. Some reviewers also acknowledged the value of the added empirical results and the broader experimental effort. However, the central theoretical concern raised most strongly by Reviewer Q6T9 remains unresolved: showing that two upper bounds are close does not imply that the corresponding losses or behaviors are close, and the proposed “rigorous” bounds are mathematically correct but largely uninformative for understanding progressive training. The reliance on assumptions of tight bounds is not convincingly justified, and the theory remains descriptive rather than explanatory. In addition, concerns about generality persist, as the evidence on modern LLM architectures is still limited or incomplete, and the claim that zero/one-layer initialization is inherently optimal lacks solid theoretical grounding.

**Reviewer Scores:**

Reviewer Q6T9 would likely maintain a score just below the acceptance threshold, as their main concern about the theoretical validity and informativeness of the analysis was not substantively resolved despite extended discussion. Reviewer 2adh would also likely keep their original marginally-below-threshold score, acknowledging the improved clarity and added experiments but remaining unconvinced about generality to modern architectures and the lack of theoretical justification for key design choices. Reviewer v2fW, while positive about the empirical contributions and practical relevance, would likely not increase their score and could potentially lean slightly downward, given that the theoretical limitations and incomplete architectural validation remain. Taken together, while the paper has clear empirical strengths, the unresolved theoretical weaknesses and questions about generality lead to a weak reject recommendation.

---

### Decision · Program_Chairs · 2026-01-26

Reject